# Non-Destructive Analysis of Chlorpheniramine Maleate Tablets and Granules by Chemometrics-Assisted Attenuated Total Reflectance Infrared Spectroscopy

**DOI:** 10.3390/molecules27123760

**Published:** 2022-06-10

**Authors:** Chutima Phechkrajang, Putthiporn Khongkaew, Waree Limwikrant, Montree Jaturanpinyo

**Affiliations:** 1Department of Pharmaceutical Chemistry, Faculty of Pharmacy, Mahidol University, Bangkok 10400, Thailand; putthiporn@go.buu.ac.th; 2Faculty of Pharmaceutical Science, Burapha University, Chonburi 20131, Thailand; 3Department of Manufacturing Pharmacy, Faculty of Pharmacy, Mahidol University, Bangkok 10400, Thailand; waree.lim@mahidol.ac.th (W.L.); montree.jat@mahidol.ac.th (M.J.)

**Keywords:** non-destructive analysis, FT-IR, content uniformity, PLSR

## Abstract

Non-destructive analysis of chlorpheniramine maleate (CPM), pharmaceutical tablets, and granules was conducted by chemometrics-assisted attenuated total reflectance infrared spectroscopy (ATR-IR). For tablets, an optimum PLSR model with eight latent factors was obtained from area-normalized and standard normal variate (SNV) pretreated ATR-IR spectral data with correlation coefficients (R^2^) of calibration and cross-validation of 0.9716 and 0.9602, respectively. The model capability for the 42 test set samples was proven with R^2^ between the reference and model prediction values of 0.9632, and a root-mean-square error of prediction (RMSEP) of 1.7786. The successive PLSR model for granules was constructed from SNV and first derivative pretreated ATR-IR spectral data with two latent factors and correlation coefficients (R^2^) of calibration and cross-validation of 0.9577 and 0.9450, respectively.

## 1. Introduction

Uniformity of the dosage unit is a quality attribute indicating consistency of pharmaceutical dosage forms. This measurement is required to ensure product quality, efficacy, and patient safety. Content uniformity (CU) is a uniform dosage unit test for narrow therapeutic index drugs, especially for tablets and hard capsules whose dose or ratio of drug substance is <25 mg or 25%. In general, CU is performed using the procedure for assay, but the analysis is conducted for 10 or 30 individual units in dosage form for CU [1]. Therefore, CU testing is usually time- and reagent-consuming.

Applications for non-destructive analysis based on chemometrics-assisted spectroscopic methods such as near-infrared (NIR) spectroscopy, Raman spectroscopy, Terahertz spectroscopy, Fourier transform infrared spectroscopy (FTIR) for process analytical technology (PAT), and quality control in pharmaceutical manufacturing are increasing [2,3,4,5,6,7,8]. In addition, trend quality guidelines for regulation use chemometrics-assisted spectroscopic methods in drug quality control [9].

Chemometrics is defined as a chemical discipline using mathematical, statistical, and other methods to accomplish objectives, e.g., the optimal measurement procedure, the optimal experiment condition, and the amount of relevant chemical information by chemical data analysis. The most commonly used chemometric technique for quantitative analysis is a supervised technique, namely partial least square regression or PLSR [1]. Chemometric-assisted spectrophotometric methods are widely used for determining active substances in pharmaceutical and herbal medicine samples [10,11].

Fourier transform infrared spectroscopy (FTIR) coupled with chemometrics for pharmaceutical analysis and quality control in the pharmaceutical industry has been reported [12,13,14,15,16]. Eid et al. (2020) reported a PLSR method for the quantitative determination of vildagliptin and metformin in pharmaceutical combinations with diverged concentration ranges [12]. Lawson et al. (2018) presented a PLSR model for the rapid identification of paracetamol in counterfeit medicines [13]. The application of FTIR and chemometrics for the quantitative determination of anti-inflammatory drugs was reported by Hassib et al. (2017) [14]. Sruthi et al. (2018) and Riyanto et al. (2014) presented chemometrics-assisted FTIR methods for determining levosulpiride and methamphetamine, respectively [15,16].

The non-destructive analysis of pharmaceutical dosage forms using FTIR coupled with chemometrics is attractive since it is fast, simple, and environmentally friendly. However, its application in the low concentration dosage has problems because IR transmittance signals from a desired active pharmaceutical ingredient (API) may interfere with major excipients in the formulation. However, many pharmaceutical products are produced as low-content tablets. These tablets must be studied separately for content uniformity (CU) using the corresponding assay method with at least 10 individual tablets. Therefore, this challenges our study to demonstrate whether ATR-IR coupled with chemometrics can be employed for low-content tablets that use chlorpheniramine maleate (CPM) as the model drug.

Quantitative determinations of CPM use high-pressure liquid chromatography (HPLC) [17,18,19], direct current polarography [20], near-infrared chemical imaging [21], and UV–vis spectrophotometric methods [22,23]. Assay methods based on solvent extraction steps prior to UV spectrophotometric measurements for tablets’ CPM content are described in the standard methods of the United State Pharmacopia (USP) 2022 [24]. Therefore, CU testing of CPM using the USP reference method is time-consuming, suffering from sample preparation steps and producing organic solvent waste.

We used chlorpheniramine maleate (CPM) tablets and granules in this study as representative samples to develop non-destructive analysis methods based on chemometrics-assisted FT-IR spectroscopy. The developed methods can be used as alternative procedures for CU testing and monitoring CPM content in the final mix granules before tableting. The candidate methods are fast, simple, and more environmentally friendly than the UV-spectrophotometric standard method [24].

## 2. Results

### 2.1. ATR-IR Measurement

CPM tablets with typical IR spectra of 4–30 mg/tablet (seven concentration levels) are shown in Figure 1A. IR spectra peaks corresponded to functional groups in the CPM molecule (Figure 1B), i.e., C=O (~1700 cm^−1^), C-H str (~2900 cm^−1^), C=N (~1640 cm^−1^), C=C (~1600 cm^−1^), C-O str (~1100 cm^−1^), and C-H bending (~880 cm^−1^). The same IR spectra were obtained for granules samples as tablets.

### 2.2. HPLC Method and Method Validation

The peak of CPM appeared at 3.1 min using the HPLC system described in Section 5.4. The CPM peak’s retention time in the sample solution was close to the retention time of the principal peak in the standard solution. As shown in Table 1, the method’s validation results were acceptable, with R^2^ values higher than 0.999 for linearity (*n* = 3). Accuracy expressed in terms of % recovery values ranged from 100.0–102.9%. Repeatability and intermediate precision, expressed in terms of RSD percentage of recovery percentage values, were 1.26 (*n* = 9) and 1.13% (*n* = 18), respectively. Specificity was approved with the peak purity index of CPM peaks from the chromatogram of standard spiked placebo and chromatogram of the standard solution. The peak purity index was close to 1.0, indicating that the pure peak of CPM was eluted without interference from other excipients.

### 2.3. PLSR Modelling

PLSR models for determining CPM in tablets were constructed from 168 IR spectra of calibration samples using HPLC values as references. As seen in Figure 1A, ATR-IR spectra of CPM had weak signals and contained noise. Data transformations are useful for reducing noise, baseline shift, and enlarging informative signals [25,26,27,28]. A total of 13 models were developed, as shown in Table 2. The optimum PLSR model was obtained from the spectra intervals 500–1700 and 2500–4000 cm^−1^ with area normalization and standard normal variate (SNV) data pretreatments. The model was constructed from eight latent factors with correlation coefficients (R^2^) of calibration and cross-validation of 0.9716 and 0.9602, respectively. Plots of calibration and cross-validation of the model for tablets are displayed in Figure 2.

By comparing the score plots of the original and pretreatment data of the best model (area-normalization and SNV), it was seen that the pretreated data were better grouped by concentration (Figure 3A) compared with the original data (Figure 3B). The model parameters such as R^2^ of model and prediction, RMSEP, and bias were superior to other models.

As shown in Table 3, 11 PLSR models were developed for the quantitative determination of CPM content in the granule samples. The optimum model was obtained from SNV and first derivative with two polynomial orders and 11 smoothing points in the pretreated data. The model was constructed from two latent factors and the spectral interval of 400–3700 cm^−1^. The model had correlation coefficients (R^2^) of calibration and cross-validation of 0.9577 and 0.9450, respectively. Calibration and cross-validation plots of the granules model are shown in Figure 4.

In the score plots of the original (Figure 5A) versus pretreated spectral data (Figure 5B) after pretreatment, the samples were clearly grouped by concentrations and PC1. The prediction ability parameters such as the model’s R^2^ of prediction, RMSEP, and bias were 0.9858, 8.0012, and −0.4014, respectively.

### 2.4. Quantitative Determination of CPM Tablets and Granules by PLSR and HPLC Methods

The 42 CPM tablets (4–30 mg/tablet) not used in PLSR modeling were used as external validation samples. The determination results obtained from the HPLC method (reference values) were plotted alongside those from the optimum PLSR model. As displayed in Figure 6A, the results from the two methods had good agreement with the correlation coefficient (R^2^ Pearson) of 0.9632. In addition, the HPLC and PLSR methods’ determination results were compared statistically. The results were not significantly different at a 95% confidence interval with *p*-value of 0.99. The residual plots of the tablet model are displayed in Figure 6B. The residuals of the data set were normally scattered, but they showed a little heteroscedasticity for higher concentrations.

Twenty-one CPM granules of test set samples were determined using the optimal PLSR model. The prediction plot for 21 granules in Figure 7A shows that the correlation coefficient (R^2^ Pearson) of 0.9858 was obtained with RMSEP and bias values of 8.0012 and −0.4014, respectively. A normal pattern of residual plots was obtained for the granules model (Figure 7B). The PLSR model and HPLC method’s determination results were also statistically compared. We found no significant difference between the concentrations of CPM in granules from the two methods (*p*-value = 0.98).

## 3. Discussion

In this study, the HPLC method reported by Sirigiri et al. [29] was used to quantitatively determine the actual concentration of CPM in all tablet and granule samples. However, the column dimension used in this study differed from that used in Sirigiri et al.’s study (3.9 × 150 mm, 5 µm versus 4.6 × 250 mm, 5 µm). From the USP general chapter <621> [30], the particle size and/or the column length can be modified for isocratic separations if the ratio of the column length (L) to the particle size (dp) remains constant or within the range of −25–50% of the prescribed L/dp ratio. The L/dp ratio limit of Sirigiri et al. [29] was 37,500–75,000. The L/dp in our study was 30,000 and exceeded the allowed limit of USP. Therefore, the HPLC procedure used in this study was validated for linearity and range, accuracy, precision, and specificity.

We demonstrated the non-destructive analysis of CPM content in tablets and granules by ATR-IR and chemometrics (PLSR). ATR mode in FTIR allowed fast, simple, and non-destructive measurement. However, its weak signals and noise is a drawback, especially in quantitative analysis. The application of IR absorption in quantitative analysis usually requires chemometrics for these reasons [14,15,16].

PLSR is a spectral decomposition technique highly used in multivariate calibration methods. The advantage of PLSR over other multivariate calibration methods, e.g., principal component regression (PCR), is that spectral data and property or assay data are used together to create a model. Property data are used to find a correlating pattern in the spectroscopic data while ensuring that the estimated regression factors are relevant to the chemical values [31,32]. In PLSR, a set of samples, namely a calibration set with spectral data and the desired property, were used to build the prediction model. Then, the prediction ability of the constructed model was determined by the desired property for a set of samples, namely the validation set or test set, that did not contribute to constructing the model. For this purpose, several calibration and validation samples were set up for the PLSR modeling of CPM tablets and granules (Table 4).

Various PLSR models were obtained for tablets and granule samples (Table 2 and Table 3). The criteria for selecting a suitable model include high R^2^ values (R^2^ model and R^2^ Pearson) and low RMSEC, RMSEP, and bias. Several models in Table 2 were found to be acceptable using these criteria, such as models 5, 8, 9, 10, 11, and 13. Model 13 was selected as the most suitable model for tablets because it has a high R^2^ model, R^2^ Pearson, and almost the lowest bias. The bias of model 13 is almost three times less than model 11. Bias is an important parameter indicating the model’s accuracy and prediction ability. Model 13 was less superior to other models for this reason.

Wavelength selection is an important factor in obtaining the appreciated model. Normally, most IR absorption bands from a molecule’s functional groups are present at wavenumbers around 500–1800 cm^−1^ and 2800–3500 cm^−1^. The R^2^ of models constructed from the selected spectral range of 500–1700 cm^−1^ or 500–1700 cm^−1^ + 2500–4000 cm^−1^ (model 5, 8–13) were superior to those obtained from the overall spectral range (model 1–4, 6–7). The models contributed by the overall spectral range usually contain simultaneously useful and useless information or noise. Therefore, the R^2^ values of those models were less than the models constructed from the selected spectral range. For the granules model, R^2^ obtained from some wavelength regions and the overall spectral range was almost the same. However, the model error in terms of RMSEC, RMSEP, and bias parameters was potentially reduced in models constructed from SNV and first derivative data pretreatment. This finding may be because data pretreatment can reduce spectral noise and enlarge informative signals.

The residual plots of prediction results obtained from the optimum models were evaluated. As seen in Figure 6B and Figure 7B, the residual plots of the tablet model showed little heteroscedasticity for higher concentrations, whereas a normal pattern of residual plots was obtained for the granules model. These results may be because the final mix granule powder was ground before the ATR-IR measurement, resulting in reducing the particle size distribution and increasing the consistency of ATR-IR measurement for the same CPM concentration level. The final mix of granule powder was compressed without prior grinding for the tablet. Therefore, the appearance of one surface component of the mixture in greater amounts than that expected from the mass ratio may occur [33] and bring about inconsistencies in the ATR-IR measurements.

## 4. Conclusions

We successfully developed non-destructive analysis methods for CPM tablets and granules with chemometrics-assisted ATR-IR. The candidate method was superior to the UV-Visible and fluorescence spectroscopy, in which the sample preparation step was not required. The samples were directly placed onto the ATR-IR instrument for spectrum measurement. In addition, high throughput analysis was allowed without producing the chemical waste. Our results showed that data transformation was required to reduce spectral noise and improve ATR-IR spectral data. The final model for the tablets was obtained from data in the wavelength intervals of 500–1700 and 2500–4000 cm^−1^ with area normalization and standard normal variate (SNV) data pretreatments. The optimal model for granules was obtained from SNV and first derivative data transformation of ATR-IR spectral data in the range of 400–3700 cm^−1^. For both tablets and granules, the PLSR models’ determination results statistically agreed with the HPLC method, indicating that ATR-IR combined with PLSR could be a fast, simple, and non-destructive alternative method for the quality control of drug substances in both in-process manufacturing and finished product control. In addition, our findings strongly support ATR-IR coupled with chemometrics for the assay of low concentration content tablets and in-process granule samples. For CU testing, our results showed that the candidate method had the potential for individual analysis of CPM tablets at 4 mg/Tablet. However, to accomplish the CU analytical concentration range of 70–130%, the ability of the method at 70% concentration or 2.8 mg/tablet should be further investigated in a future study.

## 5. Experimental

### 5.1. Chemicals and Reagents

Chlorpheniramine maleate (CPM) was purchased from S. Tong Chemicals Co., Ltd. (Nonthaburi, Thailand). Lactose monohydrate, croscarmellose sodium, and magnesium stearate were obtained from Maxway Co., Ltd. (Bangkok, Thailand). Tapioca starch and corn starch were supplied from National Starch and Chemical Co., Ltd. (Rayong, Thailand).

### 5.2. Preparation of Chlorpheniramine Maleate Tablets

As shown in Table 5, chlorpheniramine maleate (CPM) tablets with seven strengths (4, 8, 10, 15, 20, 25, and 30 mg/tablet) were prepared by the wet granulation method. CPM, lactose monohydrate, tapioca starch, and one-half of croscarmellose sodium were dry mixed in a rotomixer for 5 min. Corn starch was dispersed in water and heated to 60–70 °C. This starch paste was poured into the dry mix and mixed using a pestle and mortar until a damp mass was obtained. The damp mass was passed through sieve No. 14. The obtained granules were dried at 50 °C for 4 h and then passed through sieve No. 18. The granules were finally mixed with the remaining croscarmellose sodium for 5 min and magnesium stearate for 3 min by a rotomixer. This powder mix was ready to be tableted. The diameter of each tablet was set at 6 mm, and the average weight per tablet was 132 mg for all formulations. The prepared tablets were sampled and characterized for hardness, friability, disintegration time, and weight variation to ensure they complied with the USP standard before further analysis.

### 5.3. ATR-IR Measurement

A total of 210 tablets (30 tablets for each strength) were directly measured by an FTIR spectrophotometer (Nicolet iS5, Thermo Scientific, Waltham, MA, USA) with attenuated total reflectance (ATR) mode. The detector was deuterated triglycine sulfate (DTGS). For granules, 70 granule samples (7 strengths and 10 samples from each strength) were directly placed onto the FTIR instrument.

### 5.4. HPLC Analysis

After ATR-IR measurement, the tablets were separately assayed with a published HPLC method described by Sirigiri et al. [29]. The HPLC condition consisted of a Symmetry^®^ C18 (3.9 × 150 mm, 5 µm) column and a mobile phase mixture of water (pH 2 adjusted with orthophosphoric acid): acetonitrile (60:40 *v*/*v*). The flow rate was 1.0 mL/min, and the photodiode array detector was 218 nm. To prepare the sample solution, a tablet (or a portion of granule equivalents to one tablet) was dissolved and adjusted to 25.0 mL with the diluent. A mixture of water and acetonitrile, 50:50 (% *v*/*v*), was used as the diluent in the HPLC experiment. Then, 1.0 mL was transferred to a 10 mL volumetric flask and adjusted to the mark with diluent. The solution was filtered with a 0.45 µm syringe filter membrane before being injected into the HPLC system. The concentration of CPM in the sample was calculated using the linear equation of the calibration curve plotted between 10–30 µg/mL of CPM standard. The actual CPM content in the granule samples was acquired from the HPLC method in the same manner as the tablet’s condition.

### 5.5. PLSR Modeling

A schematic diagram for the PLSR models is illustrated in Figure 8. The IR spectra of 210 tablets containing 4–30 mg/tablet were imported into Unscrambler to construct the PLSR model. A total of 168 samples were selected by the Kennard and Stone algorithm [34] and used as calibration samples. The remaining 42 samples served as validation samples. Several pretreatment methods, such as the standard normal variate (SNV), area normalization, first (D1) and second (D2) derivatives, and the two combined pretreatment methods were applied to the original data. Various PLSR models were developed based on the original and pretreated data with respect to reference values from the HPLC method. A suitable model was selected from the optimal parameters, i.e., R^2^ of calibration model, R^2^ of cross-validation, root-mean-square error of calibration (RMSEC), root-mean-square error of prediction (RMSEP), bias, and the prediction ability of the validation samples. The RMSEP and bias were calculated from the following equations [35]:(1)$$\mathrm{RMSEP}=\sqrt{\frac{\sum_{i=1}^{n}{\left(y_{i}-y_{i,\mathrm{ref}}\right)}^{2}}{n}}$$
(2)$$\mathrm{Bias}=\frac{1}{n}\overset{n}{\underset{i=1}{\sum }}\left(y_{i}-y_{i,\mathrm{ref}}\right)\;$$
where n is number of validation samples, y_i_ is the determination value from ATR-IR, and y_i,ref_ is the determination value from the HPLC method.

The PLSR models for the granules were performed in the same manner as the tablets. Seventy elements of AIR-IR spectral data were imported into Unscrambler to construct various PLSR models from original and pretreated data. Forty-nine samples (seven strengths with seven samples from each strength) were randomly selected as calibration samples. The remaining 21 samples were used as validation samples. Various PLSR models were performed from original and pretreated data with respect to reference values from the HPLC method. A suitable model was selected from the optimal parameters, i.e., R^2^ of the calibration model, R^2^ of cross-validation, root-mean-square error of prediction (RMSEP), bias, and the prediction ability of validation samples. The calibration and validation samples’ composition for CPM tablets and granules are presented in Table 4.

## Figures and Tables

**Figure 1 molecules-27-03760-f001:**
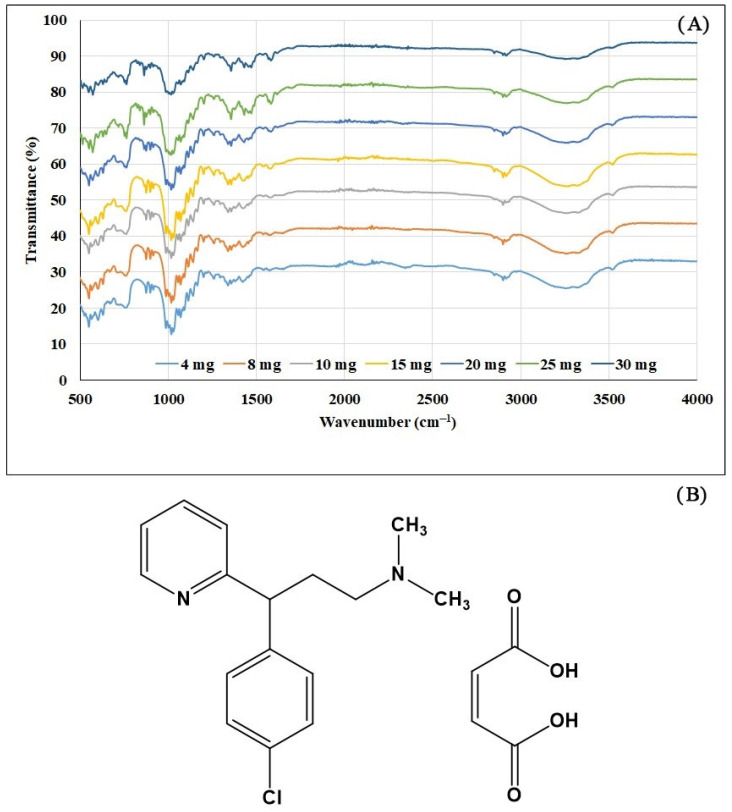
(**A**) Typical IR spectra of chlorpheniramine maleate tablets (4–30 mg/tablet). (**B**) Chemical structure of chlorpheniramine maleate.

**Figure 2 molecules-27-03760-f002:**
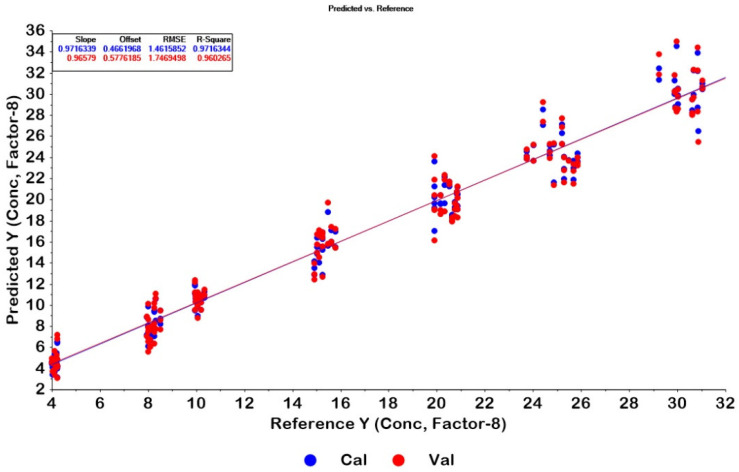
Plots of calibration and cross-validation of the optimum model for tablets. Calibration and cross-validation samples are in agreement, indicating that the data are appreciated modelled, the closer the slope is to 1, R^2^ of model (R^2^ calibration) and R^2^ validation are close together indicating a good fit of model and prediction ability for future samples.

**Figure 3 molecules-27-03760-f003:**
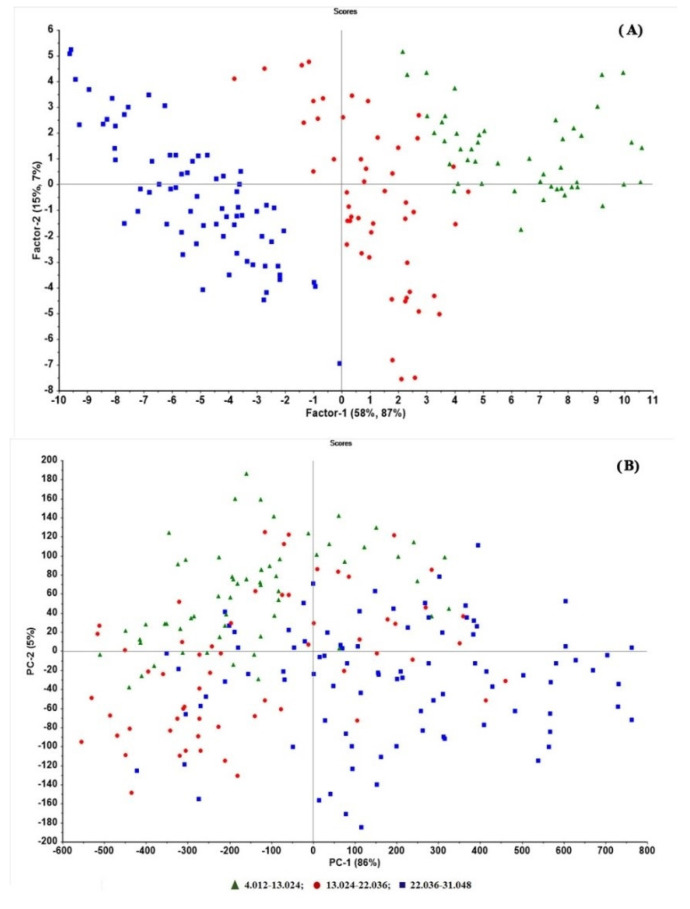
Score plots of (**A**) the pretreated data (area normalization and SNV) and (**B**) original data of the best model for tablets. The data grouping by concentrations along with PC1 was clearly seen from the pretreated data compared the original data.

**Figure 4 molecules-27-03760-f004:**
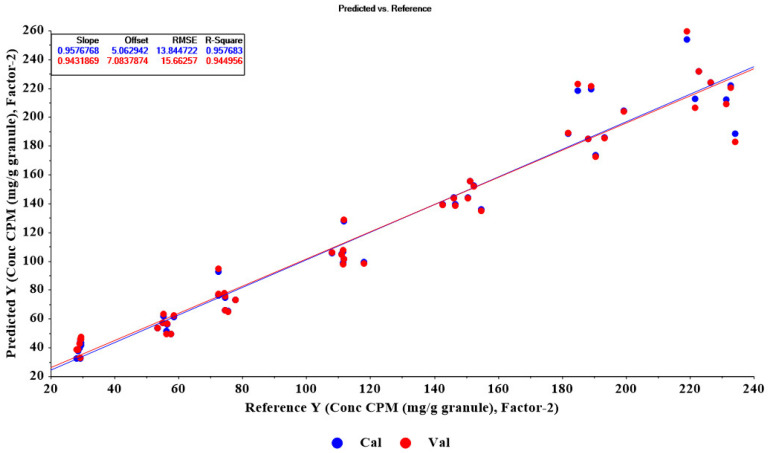
Plots of calibration and cross-validation of the optimum model for granules. Calibration and cross-validation samples indicating that the data are appreciated and modelled, the closer the slope is to 1, R^2^ of model (R^2^ calibration) and R^2^ validation are close together, indicating a good fit of model and prediction ability for future samples.

**Figure 5 molecules-27-03760-f005:**
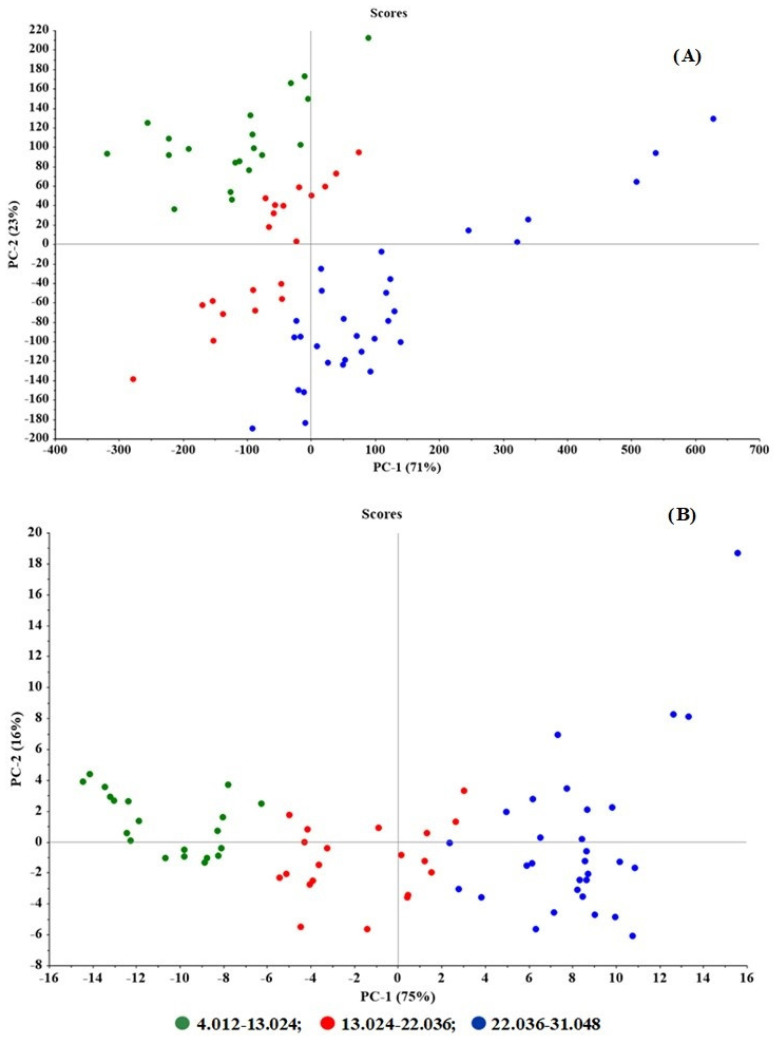
Score plots of (**A**) original data and (**B**) the pretreated data (SNV + 1st derivative) of the best model for granules. The data grouping by concentrations along with PC1 was clearly seen from the pretreated data compared the original data.

**Figure 6 molecules-27-03760-f006:**
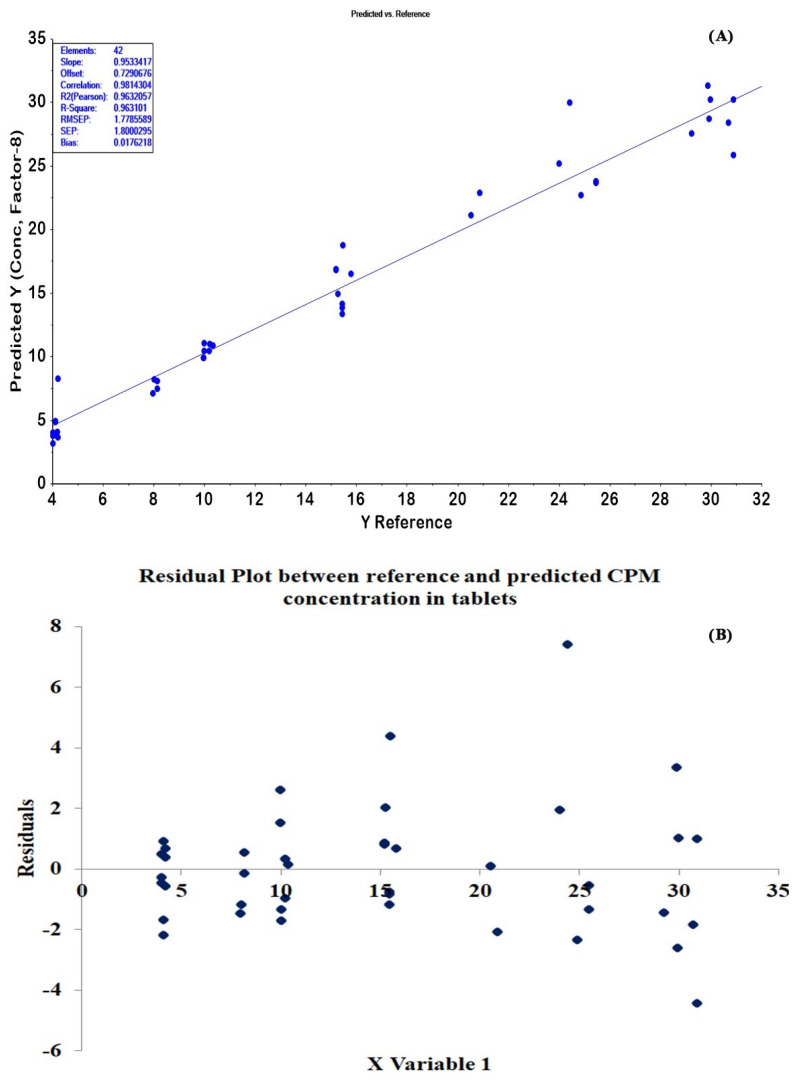
(**A**) The correlation plot between reference and predicted CPM contents of test set samples for tablets showed the slope close to 1, a good R^2^ (Pearson), low RMSEP and bias. (**B**) The residuals plots of the prediction values compared with reference values showed a little heteroscedasticity for the high concentrations.

**Figure 7 molecules-27-03760-f007:**
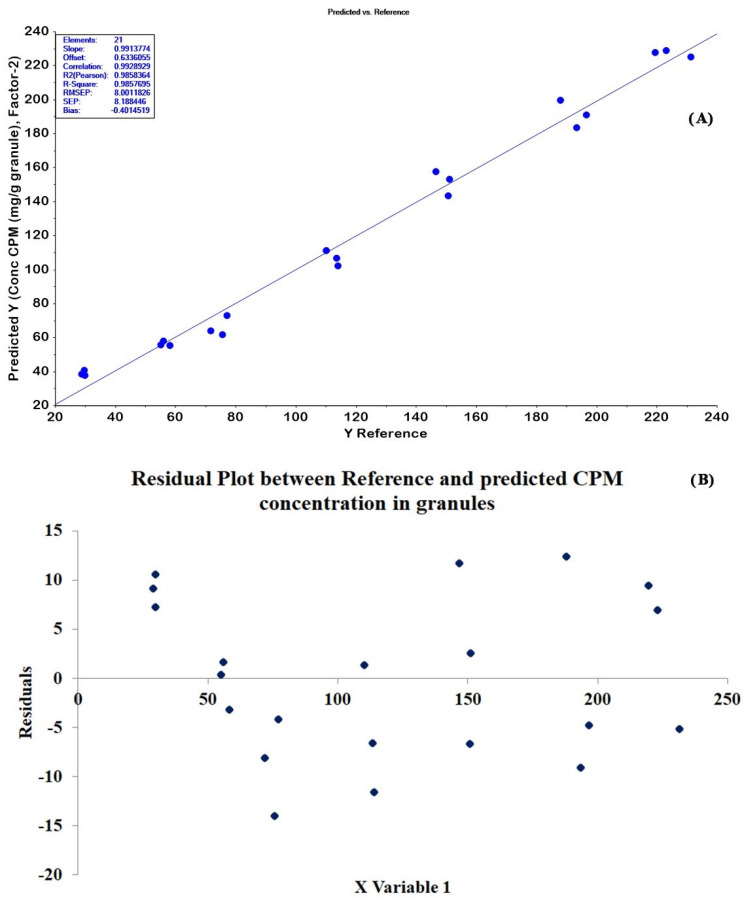
(**A**) The correlation plot between reference and predicted CPM contents of test set samples for granules showed the slope close to 1, a good R^2^ (Pearson), low RMSEP and bias. (**B**) The residuals plots of the prediction values compared with reference values showed the random distribution of the residual values with respect to reference values.

**Figure 8 molecules-27-03760-f008:**
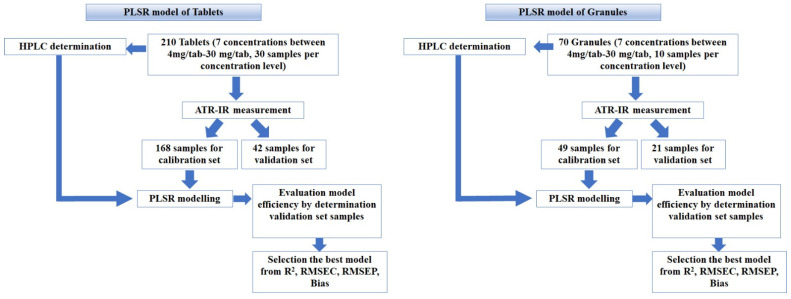
Schematic diagram for PLSR models construction for tablets and granules.

**Table 1 molecules-27-03760-t001:** HPLC method validation results.

Method Validation Characteristics	Results
Range	10–30 µg/mL
Linearity	
Equation	y = 14,603x − 2363
R^2^ (*n* = 3)	0.9999
Accuracy (% Recovery)	100.0–102.9%
Precision	
Repeatability (*n* = 9)	1.26
Intermediate precision (*n* = 18)	1.13
Specificity	
Peak purity index (standard)	1.0000
Peak purity index (standard spiked placebo)	0.9999

**Table 2 molecules-27-03760-t002:** The developed PLSR models and models parameter.

Model Number	Spectral Range (cm^−1^)	Spectral Data *	Latent Factors	R^2^ (Model)	R^2^ (Pearson)	RMSEC	RMSEP	Bias	Derivative Polynomial Order
1	400–4000	original	5	0.9142	0.9328	2.5419	2.4797	0.1986	-
2	400–4000	D2	9	0.9545	0.9309	1.8506	2.4850	−0.0359	-
3	400–4000	area-normalized	4	0.9083	0.9287	2.6284	2.5648	0.1862	-
4	400–4000	area-normalized + SNV	3	0.9355	0.9438	2.2040	2.2418	0.2695	-
5	2700–4000	area-normalized + SNV	5	0.9605	0.9591	1.7251	2.0673	−0.1105	-
6	400–4000	D1	5	0.9264	0.9312	2.3546	2.478	−0.0123	2 order 11 pt.
7	400–4000	D1 + SNV	9	0.9394	0.941	2.1359	2.3093	−0.2371	2 order 11 pt.
8	500–1700	D2	7	0.9716	0.9417	1.4625	2.2801	−0.3258	2 order 11 pt.
9	2500–4000	D2	5	0.9823	0.9579	1.1555	2.0986	−0.1973	2 order 11 pt.
10	500–1700, 2500–4000	D2	7	0.9840	0.9562	1.0986	2.0155	−0.415	2 order 11 pt.
11	500–1700	area-normalized + SNV	8	0.9741	0.9639	1.3977	1.7611	0.0495	-
12	2500–4000	area-normalized + SNV	8	0.9481	0.9508	1.9759	2.1874	−0.1130	-
13	500–1700, 2500–4000	area-normalized + SNV	8	0.9716	0.9632	1.4616	1.7786	0.0176	-

* D1 = First derivative, D2 = second derivative, original = original spectral data, SNV = standard normal variate.

**Table 3 molecules-27-03760-t003:** The developed PLSR models and model parameters of granules.

Model Number	Spectral Range (cm^−1^)	Spectral Data *	Latent Factors	R^2^ (Model)	R^2^ (Pearson)	RMSEC	RMSEP	Bias	Derivative Polynomial Order
1	400–3700	Original	2	0.9498	0.9762	15.0802	10.6003	−2.0409	-
2	400–1700, 2800–3700	Original 2	2	0.9395	0.9782	16.5568	10.1016	−1.8677	-
3	400–3700	SNV	1	0.9366	0.9835	16.9389	8.6349	−0.3478	-
4	400–1700, 2800–3700	SNV	1	0.9355	0.9838	17.0872	8.5433	−0.3631	-
5	400–3700	Area normalization	2	0.9295	0.9730	17.8691	11.2178	−2.0792	-
6	400–3700	SNV + D1	2	0.9577	0.9858	13.8447	8.0012	−0.4014	2 order 11 pt.
7	400–3700	SNV + D1	2	0.9572	0.9851	13.9199	8.4352	−0.6258	2 order 21 pt.
8	400–3700	SNV + D1	1	0.9362	0.9833	16.9942	8.6657	−0.3323	3 order 11 pt.
9	400–3700	SNV + D1	1	0.9364	0.9834	16.9678	8.6601	−0.3574	3 order 21 pt.
10	400–3700	SNV + D1	1	0.9368	0.9835	16.9125	8.6220	−0.3550	4 order 11 pt.
11	400–3700	SNV + D1	1	0.9368	0.9835	16.9196	8.6263	−0.3573	4 order 21 pt.

* D1 = First derivative, original = original spectral data, SNV = standard normal variate.

**Table 4 molecules-27-03760-t004:** Number of calibration and validation samples for building up PLSR models of tablets and granules.

Active Content (%)	Tablets Model		Granules Model	
	Calibration	Validation	Calibration	Validation
4	24	6	7	3
8	24	6	7	3
12	24	6	7	3
15	24	6	7	3
20	24	6	7	3
25	24	6	7	3
30	24	6	7	3
Total	168	42	49	21

**Table 5 molecules-27-03760-t005:** Composition of chlorpheniramine maleate (CPM) tablets.

Composition (mg/Tablet)	Formulation						
	1	2	3	4	5	6	7
CPM	4.0	8.0	10.0	15.0	20.0	25.0	30.0
Lactose monohydrate	80.0	76.0	74.0	69.0	64.0	59.0	54.0
Tapioca starch	40.0	40.0	40.0	40.0	40.0	40.0	40.0
Croscarmellose sodium	3.7	3.7	3.7	3.7	3.7	3.7	3.7
Corn starch	3.6	3.6	3.6	3.6	3.6	3.6	3.6
Magnesium stearate	0.7	0.7	0.7	0.7	0.7	0.7	0.7

## Data Availability

Not applicable.

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
