# Peer review of "Non-Destructive Analysis of Chlorpheniramine Maleate Tablets and Granules by Chemometrics-Assisted Attenuated Total Reflectance Infrared Spectroscopy"

_molecules, 2022, doi:10.3390/molecules27123760_

Round 1
Reviewer 1 Report
Manuscript molecules-1721197 authored by Dr. Chutima Phechkrajang et all is an interesting research regarding the use of ATR-IR as an PAT tool in order to perform quantitative characterization (potency) of tablets and granules with Chlorpheniramine Maleate. The manuscript might be improved considering the following aspects witch are more or less important:
- As far as I know commercially available tablets with Chlorpheniramine Maleate contain 4 mg/tablet. Authors might want to offer a proper justification for the chosen calibration range for tablets (4-30 mg CPM/tablet). My major concern is that your multivariate model won`t be able to asses CU for commercial tablets. The range for content uniformity, should cover a minimum of 70 to 130 percent of the test concentration.
- "Linearity assessment, apart from comparison of reference and predicted results, should include information on how the analytical procedure error (residuals) changes across the calibration range. Graphical plots can be used to assess the residuals of the model prediction across the working range. "
- Regarding accuracy and precision of multivariate models, please include RMSEC for multivariate models. "If RMSEP is found to be comparable to acceptable root mean-squared error of calibration (RMSEC) then this indicates that the model is accurate enough when tested with an independent test set." For tablets with high potencies the precision seems somehow low (Figure 4)
- Please provide more examples (in Introduction section) of applications of ATR-IR in quality control of finished and intermediate products in pharmaceutical industry. Only 2 examples are not enough. Also, in the same section please provide some other analytical techniques used for quality control of CPM containing pharmaceutical products.
- The preparation and composition of granules is not described. I assume that the composition of granule is the one of tablets minus magnesium stearate and minus half of the croscarmellose sodium.
- The HPLC method used in this research is not the same with the one cited (reference 17) since the length of the column is not the same and of course retention time is different.
- The calibration and validation samples for both tablets and granules should be presented in a table ( please see article https://doi.org/10.1016/j.molstruc.2021.131326 ).
- Rows 47-48 - PLSR in partial least squares regression
- Row 127 - There is a mention about figures 3A and 3B. Please correct this inconsistency.
- Row 337 - equation 1 RMSEP instead of MSEPR
Author Response
Reviewer 1
- As far as I know commercially available tablets with Chlorpheniramine Maleate contain 4 mg/tablet. Authors might want to offer a proper justification for the chosen calibration range for tablets (4-30 mg CPM/tablet). My major concern is that your multivariate model won`t be able to asses CU for commercial tablets. The range for content uniformity, should cover a minimum of 70 to 130 percent of the test concentration.
Answer: We thank the reviewer for the comment. We agree with this comment. According to ICH Q2(R2), range of analytical procedure for content uniformity should cover 70-130% of declared content. However, the main objective of our study is proving whether ATR-FTIR coupled with chemometrics is able to quantitate a tablet that falls into content uniformity testing criteria (the content in a dosage unit is equal or less than 25 mg or 25%) by using chlorpheniramine maleate (CPM) as the representative drug. Never the less, the authors still concern for this comment. We have been randomly checked for the average weight of a commercial CPM tablet that is available in Thailand. As shows in the Table below, the average weight of commercial tablet is 110 mg compared with 132 mg of average weight of tablets used in this study. The content 4 mg/tablet is 3.6 % (w/w) for commercial tablet and will be 3.0 % (w/w) for our tablet. The 70% content is 2.8 mg/tablet, this amount equal 2.5 % (w/w) for commercial tablets and 2.1% (w/w) for our tablets. Although quantitative determination of CPM at 2.1% (w/w) has been not demonstrated in our study, the concentration at 70% content of commercial product closes to the lowest concentration in our study (2.5 % VS. 3.0 %). By this information, we expect that the candidate method could has potential for content uniformity testing.
|
Item |
Commercial tablets |
Our tablets |
|
Average weight/tab |
110 mg |
132 mg |
|
CPM content (mg/tab) |
4 mg |
4 mg |
|
% w/w |
3.6 % |
3.0 % |
|
70% content (mg/tab) |
2.8 mg |
2.8 mg |
|
% w/w at 70% content |
2.6 % |
2.1 % |
- "Linearity assessment, apart from comparison of reference and predicted results, should include information on how the analytical procedure error (residuals) changes across the calibration range. Graphical plots can be used to assess the residuals of the model prediction across the working range. "
Answer: We thank the reviewer for the comment. We agree with this comment. From "Linearity assessment, apart from comparison of reference and predicted results, the residual plots of these data have been added as Figure 6 B and 7 B of the revised manuscript.
- Regarding accuracy and precision of multivariate models, please include RMSEC for multivariate models. "If RMSEP is found to be comparable to acceptable root mean-squared error of calibration (RMSEC) then this indicates that the model is accurate enough when tested with an independent test set." For tablets with high potencies the precision seems somehow low (Figure 4)
Answer: We thank the reviewer for the comment. We agree with this comment. RMSEC values have been included in Table 2 and Table 3. For the good models, it was found that RMSEC and RMSEP values were consistent.
- Please provide more examples (in Introduction section) of applications of ATR-IR in quality control of finished and intermediate products in pharmaceutical industry. Only 2 examples are not enough. Also, in the same section please provide some other analytical techniques used for quality control of CPM containing pharmaceutical products.
Answer: We thank the reviewer for the comment. We agree with this comment. The Introduction section has been revised. More applications of ATR-IR in quality control of finished and intermediate products in pharmaceutical industry and analytical techniques used for quality control of CPM containing pharmaceutical products have been added in the Introduction section (Ref 12-24).
- The preparation and composition of granules is not described. I assume that the composition of granule is the one of tablets minus magnesium stearate and minus half of the croscarmellose sodium.
Answer: We thank the reviewer for the comment. We apologize for this confusing. In fact, “granules” in our study means to the powder mix (Row 468) before tableting. Therefore, the composition of granules and tablet are the same.
- The HPLC method used in this research is not the same with the one cited (reference 17) since the length of the column is not the same and of course retention time is different.
Answer: We thank the reviewer for the comment. We agree with this comment. Reference 17 is changed to reference 29 in the revised manuscript. From USP <621>, for isocratic separations, the particle size and/or the length of the column may be modified provided that the ratio of the column length (L) to the particle size (dp) remains constant or into the range between −25% and 50% of the prescribed L/dp ratio. The limit of L/dp ratio of reference 29 was 37,500 – 75,000. Since L/dp of our study was 30,000 and exceed allowed limit of USP. So, the HPLC procedure used in this study was validated for linearity and range, accuracy, precision and specificity. The method validation results of HPLC method is in Table 1.
- The calibration and validation samples for both tablets and granules should be presented in a table (please see article https://doi.org/10.1016/j.molstruc.2021.131326 ).
Answer: We thank the reviewer for the comment. We agree with this comment. (Table 4.) Composition of calibration and validation samples for tablets and granules have been added in the revised manuscript.
- Rows 47-48 - PLSR in partial least squares regression
Answer: We thank the reviewer for the comment. We agree with this comment. Rows 47-48 have been changed to Row 44 in this revised manuscript. The sentence has been changed to “partial least square regression or PLSR”.
- Row 127 - There is a mention about figures 3A and 3B. Please correct this inconsistency.
Answer: We thank the reviewer for the comment. We agree with this comment. The mention about Figure 3A and 3B have been corrected.
- Row 337 - equation 1 RMSEP instead of MSEPR
Answer: We thank the reviewer for the comment. We agree with this comment. This error has been fixed already.

Reviewer 2 Report
1) Writing has serious language errors. In any case, it is necessary to edit and correct the English language.
2) The structure of the article is fundamentally flawed. "Materials and methods" was placed after the discussion. This should be placed after the introductory part. This needs to be edited.
3) Page 2: The author claims that 210 tablets were measured and then presents only 7 IR spectrums. So 30-30 of the 7 pills containing CPM in different quantities are the same? If so, this should be written in the description of the measurement. If the reason is different, it needs to be explained.
4) Page 2 / Figure 1: the inscription of the y-axis (absorbence or intensity) is missing.
5) Page 4: the author writes "As seen from Table 1, the optimum PLSR model was obtained from spectra 90 interval 500-1,700 and 2,500-4,000 cm-1 with area normalization and standard normal 91 variate (SNV) data pretreatments". It is not clear to me why these models are the most suitable. This definitely needs to be explained.
6) Page 5, Figure 3: the illustration inscriptions are very small, almost unreadable, they need to be enlarged.
7) Page 6, Figure 4: like the previous one, the illustration inscriptions are very small, but in addition, the points indicating the measuring points depicted also need to be enlarged.
8) For me, the main line of thought in the article is missing. To compensate for this, I propose to indicate the specific goals in the introduction and emphasize the fulfillment of the goals in the discussion.
Author Response
Reviewer 2
- Writing has serious language errors. In any case, it is necessary to edit and correct the English language.
Answer: We thank the reviewer for the comment. We agree with this comment. English is corrected throughout the manuscript. Professional English editing service is use to check for grammar, style and syntax prior the resubmission.
- The structure of the article is fundamentally flawed. "Materials and methods" was placed after the discussion. This should be placed after the introductory part. This needs to be edited.
Answer: We thank the reviewer for the comment. We apologize for the article structure. However, the authors tried to prepare the article by following the format of Molecules Journal (https://www.mdpi.com/journal/molecules/instructions#manuscript). According to that author instruction, Research manuscript sections should consist of;
- Introduction
- Results
- Discussion
- Materials and Methods
- Conclusion (Optional)
The manuscript has been prepared according to this structure.
3) Page 2: The author claims that 210 tablets were measured and then presents only 7 IR spectrums. So 30-30 of the 7 pills containing CPM in different quantities are the same? If so, this should be written in the description of the measurement. If the reason is different, it needs to be explained.
Answer: We thank the reviewer for the comment. In this study, total 210 tablets are measured IR spectra. Because there are 7 concentrations for 210 tablets (4, 8, 10, 15, 20, 25, 30 mg/tab). Only one spectrum of each concentration is presented in Figure 1A as the representative.
4) Page 2 / Figure 1: the inscription of the y-axis (absorbence or intensity) is missing.
Answer: We thank the reviewer for the comment. We agree with this comment. Figure 1 is changed to Figure 1A, the inscription of the y-axis (intensity) has been added.
5) Page 4: the author writes "As seen from Table 1, the optimum PLSR model was obtained from spectra 90 interval 500-1,700 and 2,500-4,000 cm-1 with area normalization and standard normal 91 variate (SNV) data pretreatments". It is not clear to me why these models are the most suitable. This definitely needs to be explained.
Answer: We thank the reviewer for the comment. The criteria for selection a suitable model is the model which has high R2 model, R2 Pearson and low RMSEC, RMSEP and Bias. By these criteria, several models in Table 1 were found acceptable such as model 5, 8, 9, 10, 11 and 13. But we choose model 13 as the most suitable model because it has high R2 model, R2 Pearson and almost the lowest Bias. Compared with model 11, Bias of model 13 is almost 3-times less than model 11. Because Bias is the important parameter indicated to the model accuracy and prediction ability. By this reason, the model 13 was little superior more than other models.
6) Page 5, Figure 3: the illustration inscriptions are very small, almost unreadable, they need to be enlarged.
Answer: We thank the reviewer for the comment. We agree with this comment. The inscriptions of Figure 3 have been improved in the revised manuscript.
7) Page 6, Figure 4: like the previous one, the illustration inscriptions are very small, but in addition, the points indicating the measuring points depicted also need to be enlarged.
Answer: We thank the reviewer for the comment. We agree with this comment. The inscriptions of Figure 4 have been improved and the points are enlarged in the revised manuscript.
8) For me, the main line of thought in the article is missing. To compensate for this, I propose to indicate the specific goals in the introduction and emphasize the fulfillment of the goals in the discussion.
Answer: We thank the reviewer for the comment. We agree with this comment. In the revised manuscript, the main idea and specific goals of this study have been focused in the Introduction part and Discussion part.

Reviewer 3 Report
In this manuscript, Jaturanpinyo and co-workers are reporting the development of a ATR-IR spectroscopy based non-destructive analysis method for detecting Chlopheniramine malate in tablets and granules. This analytical method seems to be useful and has a practical application. The current submitted version of the manuscript seems to be deficient with explaining and discussing experimental findings. I suggest authors to carefully revise this manuscript to deliver a coherent and informative output to the readers to understand the significance of this develop method. Therefore, I recommend a major revision.
(1). Can authors provide the structures of the analyte in the manuscript and also provide a schematic representation how this method works. This will attract more attention from the readers. The currently submitted version of the manuscript lacks graphical representations and makes manuscript slightly difficult to read.
(2). I am nor sure why authors didn’t calculate the LOD and LOQ values for this new method. These values should be included in the manuscript, and I highly recommend authors to provide relevant calculations, data charts and plots in the supporting information.
(3). Can authors be specific to discuss why this analytical method superior to certain spectroscopic methods such as fluorescence or UV-Visible. For instance, fluorescence spectroscopy-based methods are well-known for their exceptional sensitivity.
(4). The introduction of the manuscript is very poor. Authors should provide more background information and also discuss the importance of this detection method using reasonable number of reported literature/references. The current introduction of the manuscript requires a major revision.
(5). I strongly suggest authors to check their grammar and English in the submitted format as I noticed multiple inconsistencies.
(6). Authors should discuss table 1 in detail in the manuscript. Authors should provide a reasonable explanation towards the changes in R 2 values in the table 1, based on the spectral range.
(7). Can authors be specific about their sample preparation method?
(8). From the comparison of table 1 vs table 2, why does the parameters seem to be well optimized for the granules with comparison to the tablets in general? Can authors explain this effect?
(9). Figure2 and Figure 3 captions must provide more information to make it more understandable for the readers.
(10). Can authors discussed in detailed about their calibration method in the manuscript. The current manuscript does not discuss this important information in detail.
(11). Figure 4, plot data points seem to be too small and hard to see. Can authors modify the figure properly?
(12). I do not see an organized conclusion section in the manuscript.
Author Response
Reviewer 3
In this manuscript, Jaturanpinyo and co-workers are reporting the development of a ATR-IR spectroscopy based non-destructive analysis method for detecting Chlopheniramine malate in tablets and granules. This analytical method seems to be useful and has a practical application. The current submitted version of the manuscript seems to be deficient with explaining and discussing experimental findings. I suggest authors to carefully revise this manuscript to deliver a coherent and informative output to the readers to understand the significance of this develop method. Therefore, I recommend a major revision.
(1). Can authors provide the structures of the analyte in the manuscript and also provide a schematic representation how this method works. This will attract more attention from the readers. The currently submitted version of the manuscript lacks graphical representations and makes manuscript slightly difficult to read.
Answer: We thank the reviewer for the comment. We agree with this comment. The structure of the analyte (chlorpheniramine maleate) has been added as Figure 1 B. In addition, a schematic representation diagram of the study has been added as Figure 8 of the revised manuscript.
(2). I am nor sure why authors didn’t calculate the LOD and LOQ values for this new method. These values should be included in the manuscript, and I highly recommend authors to provide relevant calculations, data charts and plots in the supporting information.
Answer: We thank the reviewer for the comment. For LOD and LOQ, we did not calculate and include in the article because this procedure was developed for quantitative determination of drug content in tablets and granules. This kind of method falls into categories I according to USP general chapter <1225> which is not require LOD and LOQ. The relevant data and plots have been added in the main text of the revised manuscript already.
(3). Can authors be specific to discuss why this analytical method superior to certain spectroscopic methods such as fluorescence or UV-Visible. For instance, fluorescence spectroscopy-based methods are well-known for their exceptional sensitivity.
Answer: We thank the reviewer for the comment. We agree with this comment. The advantages of ATR-IR coupled with chemometrics have been added in the conclusion section. The candidate method in this study is non-destructive analysis of CPM in tablets and granules by ATR-IR coupled with chemometrics method. This method is superior more than UV-Visible and fluorescence spectroscopy in which the sample preparation step is not required. Both tablet and granule can be directly placed onto ATR-IR instrument for spectrum measurement. In addition, this method allows for high throughput analysis and does not produce chemical waste.
(4). The introduction of the manuscript is very poor. Authors should provide more background information and also discuss the importance of this detection method using reasonable number of reported literature/references. The current introduction of the manuscript requires a major revision.
Answer: We thank the reviewer for the comment. We agree with this comment. The Introduction section has revised, more back information and important of the candidate method have been added along with reported references.
(5). I strongly suggest authors to check their grammar and English in the submitted format as I noticed multiple inconsistencies.
Answer: We thank the reviewer for the comment. We agree with this comment. English is corrected throughout the manuscript. Professional English editing service is use to check for grammar, style and syntax prior the resubmission.
(6). Authors should discuss table 1 in detail in the manuscript. Authors should provide a reasonable explanation towards the changes in R 2 values in the table 1, based on the spectral range.
Answer: We thank the reviewer for the comment. We agree with this comment. The change of R 2 values in the table 1, based on the spectral range has been added in the revised manuscript (Row 405-425). Wavelength selection is an important factor to contain the appreciated model. Normally, majority of IR absorption bands contributed from the functional groups in a molecule including CPM will present in the wavenumber around 500-1800 cm-1 and 2800-3500 cm-1. It can be seen that R2 of models constructed from the selected spectral range 500-1700 cm-1 or 500-1700 cm-1 + 2500-4000 cm-1 (model 5, 8-13) were superior more than those obtained from the overall spectral range (model 1-4, 6-7). In general, the models contributed from the overall spectral range will contain both useful and useless information or noise. Therefore, R2 model values of those models were less than the models constructed from the selected spectral range. For models of granules, R2 of the models obtained from some wavelength regions and the overall spectral range were almost the same. But the model error in terms of RMSEP and Bias parameters were potential reduced from the models constructed from SNV and first derivative data pretreatment. This may be the fact that data pretreatment can be reduced spectral noise and enlarged informative signals.
(7). Can authors be specific about their sample preparation method?
Answer: We thank the reviewer for the comment. We agree with this comment. Sample preparation procedure for HPLC analysis has been added in the revised manuscript (Row 484-493). One tablet (or a portion of granule equivalents to one tablet) was dissolved and adjusted to 25.0 mL with the diluent. The mixture of water and acetonitrile, 50:50 (% v/v) was used as the diluent through the HPLC experiment. Then, 1.0 mL was transferred to 10.0-mL volumetric flask and adjusted to the mark with diluent. The solution was filtered with 0.45µm syringe filter membrane before injection into HPLC system. For ATR-IR measurement, a tablet or a portion of granule was directly placed onto the instrument and measured without sample preparation step.
(8). From the comparison of table 1 vs table 2, why does the parameters seem to be well optimized for the granules with comparison to the tablets in general? Can authors explain this effect?
Answer: We thank the reviewer for the comment. We agree with this comment. In this study, the composition of final mix granule powder and tablet are the same. The final mix granule powder was ground before ATR-IR measurement. This can reduce particle size distribution and increase consistency of ATR-IR measurement for the same CPM concentration level. While the tablet, final mix granule powder was compressed without prior grinding. So, the appearance of one component on the surface of the mixture in greater amounts than that expected from the mass ratio may occur [reference 33] and bring about inconsistency of ATR-IR measurement.
(9). Figure 2 and Figure 3 captions must provide more information to make it more understandable for the readers.
Answer: We thank the reviewer for the comment. We agree with this comment. The captions of Figure 2-7 have been provided more in the revised manuscript.
(10). Can authors discussed in detailed about their calibration method in the manuscript. The current manuscript does not discuss this important information in detail.
Answer: We thank the reviewer for the comment. We agree with this comment. PLSR is a multivariate calibration method. The detail of PLSR has been added in the revised manuscript (Row 358-363).
(11). Figure 4, plot data points seem to be too small and hard to see. Can authors modify the figure properly?
Answer: We thank the reviewer for the comment. We agree with this comment. In the revised manuscript, Figure 4 has been changed to Figure 6A, the data points in Figure 6A have been enlarged.
(12). I do not see an organized conclusion section in the manuscript.
Answer: We thank the reviewer for the comment. We agree with this comment. Conclusion section has been added in the revised manuscript.

Reviewer 4 Report
Chemometric tests are a relatively young field of science. The use of statistical research in chemistry is very important. Matching the analytical methods to the evaluation of highly compiled substrates is very important. This research relates to a specific gap in the field. These studies show the possibility of statistical studies, I compare different analytical methods in order to choose the fastest and easiest to evaluate tablets. The description of the preparation of tablets is well described and the possibility of repeating this technology is possible. Fourier transform-infrared spectroscopy is written correctly. Perhaps on line 38 the entire name should be entered in capital letter.
HPLC analysis and PLSR modeling are described correctly and legible. There is practically no discussion of the results and what is in chapter 3 of the discussion should be included as a summary. New scientific reports were used, 2 publications are from 2022 and most of the last 10 years. I have no comments about the tables and figure, they result directly from statistical calculation.
As far as I am concerned the manuscript is well written. The subject area of research is significant in knowledge development. The introduction is interesting and correct.
Discussion of the results was carried out correctly, although there are no references to similar studies.
I have editorial comments:
l.51-55 - too small line spacing
l.56 - no space before new chapter
l.174; l.217; l.257 - too large line spacing in the description below the figure
l.181 - no space before new chapter
l.263 - spaces are missing in front of the unit
Author Response
Reviewer 4
Chemometric tests are a relatively young field of science. The use of statistical research in chemistry is very important. Matching the analytical methods to the evaluation of highly compiled substrates is very important. This research relates to a specific gap in the field. These studies show the possibility of statistical studies, I compare different analytical methods in order to choose the fastest and easiest to evaluate tablets. The description of the preparation of tablets is well described and the possibility of repeating this technology is possible. Fourier transform-infrared spectroscopy is written correctly.
Answer: The authors would like to thank reviewer for giving the valuable comments to improve the quality of our study. The authors attempted to response all comments as follow,
Perhaps on line 38 the entire name should be entered in capital letter.
Answer: We thank the reviewer for the comment. We agree with this comment. The entire name was changed to capital letter (Row 35-36 of the revised manuscript).
HPLC analysis and PLSR modeling are described correctly and legible. There is practically no discussion of the results and what is in chapter 3 of the discussion should be included as a summary. New scientific reports were used, 2 publications are from 2022 and most of the last 10 years. I have no comments about the tables and figure, they result directly from statistical calculation.
Answer: We thank the reviewer for the comment. We agree with this comment. New scientific publications have been added in the revised manuscript (Ref. 12-19, 31-33).
As far as I am concerned the manuscript is well written. The subject area of research is significant in knowledge development. The introduction is interesting and correct.
Answer: The authors would like to thank for this comment.
Discussion of the results was carried out correctly, although there are no references to similar studies.
Answer: We thank the reviewer for the comment. The references of similar studies have been added to the discussion section of the revised manuscript (References 14-16, Row 360-361).
I have editorial comments:
l.51-55 - too small line spacing
l.56 - no space before new chapter
l.174; l.217; l.257 - too large line spacing in the description below the figure
l.181 - no space before new chapter
l.263 - spaces are missing in front of the unit
Answer: We thank the reviewer for the comment. All editorial comments have been corrected already.

Round 2
Reviewer 1 Report
Dear authors,
You have carefully answered to all my questions/requests except the first observation regarding the range (4-30 mg CPM/tablet). I agree with the provided answer but still commercial tablets and the tablets included in your study have different composition (excipients mainly). My advise is to write a small phrase and present this aspect as a limitation of the study.
Author Response
Reviewer 1 (Round 2)
Comments and Suggestions for Authors
Dear authors,
You have carefully answered to all my questions/requests except the first observation regarding the range (4-30 mg CPM/tablet). I agree with the provided answer but still commercial tablets and the tablets included in your study have different composition (excipients mainly). My advise is to write a small phrase and present this aspect as a limitation of the study.
Answer: We thank the reviewer for the comment. We agree with this comment. A few sentences for this limitation are added in Row 458-461 of the manuscript revision 2. .. For CU testing, our results showed that the candidate method had the potential for individual analysis of CPM tablets at 4 mg/tab. However, to accomplish the CU analytical concentration range of 70-130%, the ability of the method at 70% concentration or 2.8 mg/tab should be further investigated in the future study.

Reviewer 2 Report
The authors have made efforts to correct the article based on the corrections I have requested. I find the article suitable for publication in the journal of Molecules.
Author Response
Reviewer 2 (Round 2)
Comments and Suggestions for Authors
The authors have made efforts to correct the article based on the corrections I have requested. I find the article suitable for publication in the journal of Molecules.
Answer: We thank the reviewer for the comment.

Reviewer 3 Report
Recommend to accept in the present form.
Author Response
Reviewer 3 (Round 2)
Comments and Suggestions for Authors
Recommend to accept in the present form.
Answer: We thank the reviewer for accept our article.
